# The Free Radical Scavenging Property of the Leaves, Branches, and Roots of *Mansoa hirsuta* DC: In Vitro Assessment, 3D Pharmacophore, and Molecular Docking Study

**DOI:** 10.3390/molecules27186016

**Published:** 2022-09-15

**Authors:** Patrícia e Silva Alves, Gagan Preet, Leandro Dias, Maria Oliveira, Rafael Silva, Isione Castro, Giovanna Silva, Joaquim Júnior, Nerilson Lima, Dulce Helena Silva, Teresinha Andrade, Marcel Jaspars, Chistiane Feitosa

**Affiliations:** 1Post-Graduation Department in Chemistry, Federal University of Piauí, Teresina 64000-040, Brazil; 2Marine Biodiscovery Centre, Department of Chemistry, University of Aberdeen, Aberdeen AB24 3UE, UK; 3Nucleus of Applied Research to Sciences (NIAC), Federal Institute of Maranhão, Campus Presidente Dutra, Presidente Dutra 65630-000, Brazil; 4Institute of Chemistry, São Paulo State University, Araraquara 14800-900, Brazil; 5Department of Pharmaceutical Sciences, Federal University of Piauí, Teresina 64049-550, Brazil; 6Department of Chemistry, Federal Institute of Piaui, Teresina 64000-040, Brazil; 7Chemistry Institute, Campus Samambaia, Federal University of Goias, Goiania 74690-900, Brazil

**Keywords:** *Mansoa hirsuta*, antioxidant, oxidative stress, triterpenes, pharmacophore, molecular docking

## Abstract

In this work, a metabolic profile of *Mansoa hirsuta* was investigated, and in vitro assays and theoretical approaches were carried out to evaluate its antioxidant potential. The phytochemical screening detected saponins, organic acids, phenols, tannins, flavonoids, and alkaloids in extracts of leaves, branches, and roots. Through LC-MS analysis, the triterpenes oleanolic acid (*m*/*z* 455 [M-H]^−^) and ursolic acid (*m*/*z* 455 [M-H]^−^) were identified as the main bioactive components. The extracts of the leaves, branches, and roots revealed moderate antioxidant potential in the DPPH test and all extracts were more active in the ABTS test. The leaf extracts showed better antioxidant capacity, displaying IC_50_ values of 43.5 ± 0.14, 63.6 ± 0.54, and 56.1 ± 0.05 µg mL^−1^ for DPPH, ABTS, and kinetics assays, respectively. The leaf extract showed higher total flavonoid content (TFC) (5.12 ± 1.02 mg QR/g), followed by branches (3.16 ± 0.88 QR/g) and roots (2.04 ± 0.52 QR/g/g). The extract of the branches exhibited higher total phenolic content (TPC) (1.07 ± 0.77 GAE/g), followed by leaves (0.58 ± 0.30 GAE/g) and roots (0.19 ± 0.47 GAE/g). Pharmacophore and molecular docking analysis were performed in order to better understand the potential mechanism of the antioxidant activity of its major metabolites.

## 1. Introduction

In recent years, a marked increase has been observed in the evaluation of naturally occurring bioactive compounds in the search for new drugs to counter oxidative stress [1,2]. Antioxidants eliminate, neutralize, or block free radicals and reactive oxygen species (ROS) present in the human body. Hence, they may prevent or delay the evolution of chronic diseases or oxidative damage caused by inhibiting oxidative stress and free radicals that attack healthy cells and tissues [1,2,3,4].

Oxidative stress plays an essential role in the pathogenesis of chronic diseases such as cardiovascular disease, diabetes, neurodegenerative diseases, and cancer. Oxidative processes are fundamental metabolic processes for all living organisms [5], but free radicals produced during chain reactions in the oxidation process are responsible for cellular damage. The imbalance of the antioxidant system and free radical damage may lead to the development of disease [6]. This oxidative effect causes damage to tissues and some biomolecules, including lipids, DNA, and proteins. Research evidence suggests that natural compounds can reduce oxidative stress and improve immune function [7]. Plants have gained considerable interest in the management of diseases related to oxidative stress due to their ability to treat diseases, since they have phytochemicals that have antioxidant properties [8]. Furthermore, skin is quite vulnerable to oxidative stress given its persistent exposure to direct ultraviolet (UV) from sunlight radiation which can bring about hyperpigmentation and pre-mature aging [9,10]. Thus, there is considerable interdisciplinary research interest in investigating new potent and effective natural and synthetic antioxidants to prevent the damage and toxicity caused by free radicals.

The skin pigment melanin plays a critical role in skin protection against induced damage by UV and free radicals. Tyrosinase is the rate-limiting enzyme in the first two steps of melanogenesis. It catalyzes the hydroxylation of L-tyrosine into L-DOPA, and oxidation of L-DOPA to form the respective O-dopaquinone [11,12].

Different skin disorders (e.g., melasma, age spots, freckles, solar lentigines and hyperpigmentation) resulted from the abnormal accumulation of melanin. Therefore, tyrosinase inhibitors are essential to ensure a decrease in the content of melanin and to design and develop new depigmenting compounds useful in pharmaceutical areas [11,12]. Antioxidants are widely used to block, delay, or enucleate oxidative stress in the human body. They are classified based on their origin as either endogenous or exogenous agents. Antioxidants act by scavenging free radicals, chelating metals, and inhibiting pro-oxidant enzyme inhibition [13]. Antioxidants give up electrons to free radicals, thereby neutralizing them [13]. The natural physicochemical and biopharmaceutical characteristics of exogenous antioxidants, including how they access target sites, play an essential role in their efficacy against oxidative stress. Hence, much attention has been paid to creating synthetic antioxidants to prevent the cellular damage caused by free radicals [14].

In this context, the search for new herbal medicines with therapeutic action included the study of *Mansoa hirsuta* DC. (Bignoniaceae), an edible plant known as garlic vine in Brazil and endemic to the semiarid region of Brazil, which stands out for its popularity and potential bioactive phytochemicals and/or functional foods. In traditional medicine, its leaves are used to control diabetes and to treat sore throats [15,16]. Studies have shown that this plant presents chemopreventive and anti-inflammatory activities by inhibiting COX, NF-κB and TNF-α [17,18], in addition to antihypertensive [19], and antifungal potential [20].

However, there are few reports of experiments carried out with methanol extracts of *M. hirsuta* leaves (EMFMh), methanol extracts of *M. hirsuta* branches (EMGMh) and methanol extracts of *M. hirsuta* roots (EMRMh) in relation to their antioxidant properties. Hence, we evaluated their antioxidant properties through 1,1-diphenyl-2-picrylhydrazyl (DPPH) and 2,2′-azino-bis-3-ethylbenzothiazoline-6-sulfonic acid (ABTS) radical scavenging methods, determination of total phenols, total flavonoids, and chemical kinetics. In addition, chemical profiles by chromatographic techniques (Thin Layer Chromatography: TLC, LC-MS, and HPLC-PDA) of the EMFMh, EMGMh and EMGMh, respectively, were determined. The correlation between the chemical profile and antioxidant activity could contribute to making *M. hirsuta* extract or bioactive compounds viable as a therapeutic strategy for the treatment of oxidative stress-related diseases with a focus on the development of safer and cheaper natural antioxidants. Also, the objective of the present study was to evaluate the antioxidant activities of the triterpenes oleanolic acid (*m*/*z* 455 [M-H]^−^) and ursolic acid (*m*/*z* 455 [M-H]^−^) were identified as the main bioactive components from *M. hirsuta*. We then used molecular docking to fit the oleanolic acid and ursolic acid into the active site of a target enzyme to identify possible correlations between the binding models and their antioxidant activities.

## 2. Results and Discussion

### 2.1. Evaluation of the Metabolic Content of Mansoa hirsuta

Phytochemicals are produced by specific biochemical pathways that occur within plant cells and are used for plant defense and adaptation to environmental stress [21]. Phytochemicals such as alkaloids, flavonoids, tannins, and phenolic acids show remarkable biological properties such as radical scavenging activity which is key to redox balance and maintenance of biological systems healthy conditions [22].

Our findings showed that the preliminary phytochemical profiling of the EMFMh, EMGMh, and EMRMh exhibited a significant array of secondary metabolites, as evidenced by the change in color or precipitation (Table 1). Similar results were also found by Raju et al. [23] and Choudhury et al. [24], in which the crude extract and fractions obtained from *M. hirsuta* leaves presented saponins, steroids, triterpenoids, phenols, tannins, anthocyanins, anthocyanidins, flavonoids (flavonol, flavone and flavanones) and/or xanthones and their heteroside derivatives. In addition, metabolites including naphthoquinones [25,26], lignans, and triterpenes [27] have also been reported.

These compounds have been commonly reported from other Bignoniaceae species, confirming this liana family as an important source of pharmaceutical bioproducts [23,24]. For example, phytochemical studies of the species *Mansoa alliacea* have shown compounds such as triterpenoids, flavonoids, and organosulfur compounds [28].

### 2.2. Chromatographic Profile of Mansoa hirsuta

#### 2.2.1. Chromatographic Analysis

TLC analysis of the reaction with Ceric Sulfate showed purple spots for all the analyzed extracts. According to Rogers and Stevens [29], monoterpenes, triterpenes and steroids usually appear as blue, purple, or gray spots. In addition, the TLC also showed low intensity spots that ranged from yellow to orange, indicating the presence of flavonoids [30]. HPLC has been proven as an efficient technique for the evaluation of complex samples from *M. hirsuta*. The chromatographic profiling of the extracts EMFMh, EMGMh and EMRMh provided relevant information concerning their metabolic composition and polarity, enabling comparison of different parts of the plant.

The screening of EMFMh, EMGMh, and EMRMh by HPLC-PDA was carried out with detection at 254 nm, 280 nm, and 366 nm. The presence of medium and high-intensity peaks with retention times in the range of 12 min to 26 min, approximately, was verified, which suggested the presence of flavonoids, proanthocyanidins, tannins, isoflavones, flavanones and dihydroflavonols by comparison with characteristic wavelengths of each class of secondary metabolites (Table 2 and Appendix A).

Flavonoids were detected from UV-Visible data in the range of 200 nm to 334 nm for the peaks with retention times of 16.12 min, 17.80 min, and 15.05 min for leaves, branches, and roots samples. According to Pachu [35], the presence of two well-defined bands at 240–285 nm (band II—ring A) and 300–400 nm (band I—ring B) are indicative of the flavonoid class (Appendix A). Such results corroborate those from the study carried out by Rocha [20] through tests for phytochemical prospection using silica gel TLC, which demonstrated the presence of flavonoids and other polyphenols in the extract of *M. hirsuta*.

Furthermore, UV-Vis spectra with maximum absorptions at wavelengths above 250 nm suggest the presence of aromatic systems [36], which are consistent with the presence of phenolic compounds in free or complexed forms to sugars and proteins. Among them, flavonoids, phenolic acids, tannins, and tocopherols stand out as some of the most common phenolic antioxidants from natural sources [37].

In the chromatographic profiles obtained at λ 254 nm, 280 nm, and 366 nm of the EMFMh, EMGMh and EMRMh, an enlarged band in the retention time ranging from 12 to 26 min was detected. According to Queiroz [38], this may be a feature associated to the presence of polymeric polyphenols (tannins). Furthermore, both extracts from leaves, branches, and roots showed peaks in the time range between 21 and 23 min which, according to Tian et al. [34], might indicate the presence of pentacyclic triterpenes oleanolic acid and ursolic acid.

The comparison of additional UV data from the main peaks in the chromatograms of the analyzed extracts showed absorption maximums close to 202–260 nm (Table 2 and Appendix A), which may be associated with proanthocyanidins (absorption in the region of 278 nm). Previous work by Svedstrom et al. [33] demonstrated the presence of oligomeric proanthocyanidins associated with absorption maxima at 235 and 280 nm. These results are also in agreement with Kamiya et al. [39], in which they identified proanthocyanidins λmax 209, 225, and 280 nm.

#### 2.2.2. LC-MS Profiling

Extracts from leaves, branches, and roots of *M. hirsuta* were analyzed using LC-MS in positive and negative modes, with peak identification performed by comparing retention times (R_t_) and mass spectral data with reference standards, given literature and database, as shown in (Appendix A) and Table 3.

The analyses disclosed the great complexity of the extracts, revealing a high number of chemical constituents. Most of the compounds detected in the EMFMh presented retention times ranging between 7.9 min and 45 min. The EMGMh disclosed substances with retention times between 2.9 min and 45 min. While the EMRMh exhibited compounds with retention times between 3.2 min and 45 min. All the analyzed extracts showed both polar and non-polar constituents.

Among the metabolites identified in this study, the presence of the pentacyclic triterpenes oleanolic acid and ursolic acid was detected in EMFMh, EMGMh and EMRMh, which were verified through comparison of molecular weights, empirical formula, and MS/MS data.

In the EMFMh (negative ionization mode), the compound with a retention time of 43.9 min presented a fragmentation profile, as described in Table 3, as well as other characteristics of these pentacyclic triterpenes, including fragments at *m*/*z* 363 (4.4%), 407 (3.4%), *m*/*z* 455 (0.8%). These MS/MS data were similar for the EMGMh (negative mode), however, in the retention time of 45.6 min with *m*/*z* 363 (19.4%), 391 (5.6%) and *m*/*z* 407 (16.7%). For EMRMh, these ions were detected, however, for a retention time of 27 min, which include *m*/*z* 363 (29.1%), 407 (16.7%), *m*/*z* 455 (5.8%). The fragments (Figure 1) use data described in the literature by Zhao et al. [40], Chen et al. [41].

#### 2.2.3. H-NMR

^1^H-NMR spectra of the EMFMh, EMGMh and EMRMh exhibited different profiles (Appendix A), for example, only the leaf extract EMFMh showed signals at the 8.0 ppm region, which is indicative of aromatic hydrogens. Low intensity singlets were observed around 9.0 ppm, possibly due to nitrogen-bonded hydrogens from aromatic alkaloids [42]. Furthermore, the EMFMh also showed signals in the range of 6–7 ppm, which, according to Lima et al. [43], may indicate the presence of α-oxygenated aromatic hydrogens.

The various detected signs of alpha-oxygenated aromatic hydrogens may indicate phenolic acids, coumarins, benzopyrones, chlorogenic acids, coumaric acids, tannins, and other phenols [43,44].

Only the EMFMh indicated signals near 5.5 ppm. Multiplets in this region 5.0–5.7 ppm can be attributed to olefin hydrogens [45]. Examples of olefinic compounds include terpenes [46], saponins [47], steroids [48], or unsaturated fatty acids [49]. Furthermore, hydrogens for aromatic compounds can be detected around 6–8 ppm. These extracts showed terpenes and saponins, as indicated by the observed signals between 0.8 and 2.9 ppm, and other compounds such as chromenes, and phytol (diterpene alcohol) [50].

The terpenes were previously reported by Sousa et al. [51] from extracts of the leaves and branches of *M. hirsuta*. Vilhema-Potiguara et al. [52], analyzed *Mansoa standleyi* extracts and reported the metabolites class triterpenoids, flavonoids, naphthoquinones, and amino acids. Silva [53] identified the pentacyclic triterpenes ursolic acid and oleanolic acid from the ethyl acetate fraction of *M. hirsuta* leaves.

In the current study, only the methanol extract of *M. hirsuta* roots showed signals at 3.0 and 4.5 ppm, which may be due to the high concentration of free sugars and heterosides. Similar spectroscopic results were reported by Munikishore et al. [54] to free sugars and heterosides, such as glycosylated flavonoids. Free sugars and heterosides can be easily detected by analyzing their ^1^H NMR spectrum, indicated by signals frequently appearing as multiplets between 3.0 and 4.5 ppm. Such NMR data were consistent with the chemosystematics data of *M. hirsuta*.

### 2.3. Antioxidant Activity by DPPH and ABTS Tests

The antioxidant capacity of plant samples is influenced by several factors such as extraction solvent and assessment method. Therefore, it is necessary to carry out different evaluations of the radical scavenging capacity and mechanisms of action. The use of more than one method to assess the antioxidant capacity of plant materials is essential due to the complex nature of chemical profiles and phytochemicals bioactivities [21]. The antioxidant activity of plant extracts can be quantified by different methods, and it is recommended to use at least two different methods [55]. The results of these tests can be used to establish a classification more precisely [56].

The antioxidant activity (AA%) of quercetin against the DPPH^•^ radical, with an IC_50_ of 41.0 μg mL^−1^ was considered as a reference value and as a positive comparative control to the antioxidant activity of methanol extracts of *M. hirsuta* leaves, branches, and roots Similar DPPH^•^ radical scavenging potential was observed for all tested samples. The EMGMh sample showed the lowest IC_50_ (77.3 ± 0.03 μg mL^−1^), meaning the highest antiradical potential, followed by EMFMh (IC_50_ 78.9 ± 0.05 μg mL^−1^) and EMRMh (IC_50_ 82.7 ± 0.11 μg mL^−1^) (Table 4).

The IC_50_ values for the antioxidant assay with ABTS^•+^ are presented in (Table 4). The extract with the highest antioxidative capacity was EMFMh (IC_50_ of 43.5 ± 0.14 µg mL^−1^), followed for EMRMh and EMGMh which presented IC_50_ equal to (56.1 ± 0.05 and 63.6 ± 0.54 µg mL^−1^), respectively.

Pereira et al. [15] reported the DPPH scavenging activity in crude ethanol extract of *M. hirsuta* leaves, with concentrations higher than (50, 100, 200 and 300 µg mL^−1^), obtained results (EC_50_ 57.1 ± 5.6 µg mL^−1^). They also performed ABTS for crude ethanol extract of leaves with concentrations of (2, 6, 12, 5, 25, 50 and 100 µg mL^−1^), and presented a (EC_50_ 14.9 ± 1.4 µg mL^−1^); however, the authors did not reported studies with other parts of this plant.

Antioxidant activity (AA) varies in medicinal plants when comparing the proposed study with the literature, and this can be attributed to particularities of each plant (such as variety, environmental conditions, harvesting methods, post-harvest treatment, and processing) and composition and concentration of the antioxidants present. For the proper determination of the antioxidant capacity, the extraction technique, its conditions, solvent used, and particular test methodology are important [57]. Other factors such as seasonal differences can directly influence the chemical constitution of the plant [58]. The plants used in other studies were collected in the Northeast of Bahia (Brazil), while our tests were carried out with plants from the North of Piauí, Brazil.

Furthermore, when compared with other species of the genus *Mansoa*, the study of Chirunthorn et al. [59] shows that the antioxidant effect is predominant in these species. Studies report DPPH antioxidant activity for petroleum ether and ethanol extract from the leaves of *Mansoa hymenaea* (19.0 and 65.7 µg mL^−1^, respectively). Da Silva et al. [60] performed tests with DDPH for methanol extracts of *M. difficilis* leaves and obtained and IC_50_ of 185.03 ± 5.45 µg mL^−1^, a value considered to be low antioxidant potential.

According to Kurt et al. [61], compounds with EC_50_ lower at concentrations below 50 µg mL^−1^ indicate high antioxidant properties, while values ranging from 50–100 µg mL^−1^, 100–200 µg mL^−1^ to above 200 µg mL^−1^ indicate moderate, low, and lack of antioxidant activity, respectively. Therefore, the extracts were more sensitive to the antioxidative test by ABTS, indicating a high antioxidant capacity for EMFMh and moderate capacity for EMRMh and EMGMh.

The greater sensitivity of extracts in the ABTS test compared to the DPPH test lies in the fact that the extracts have a greater antioxidant capacity in the elimination of the ABTS radical. According to the literature by Lee et al. [62], the ABTS assay is more sensitive to identify antioxidant activity, as it has faster reaction kinetics and a higher response to antioxidants.

The antioxidant capacities of the extracts can be attributed to the presence of phenolic compounds, such as flavonoids, tannins and lignins, which are found in plants, and which act as efficient antioxidants [63].

### 2.4. Chemical Kinetics of DPPH and ABTS

Kinetic studies help to explain how antioxidants work and allows prediction of the behavior of these substances. Using readings obtained during the determination of the antioxidant capacity in vitro, the kinetic curve of each sample’s antioxidants can be traced and, based on that, their effectiveness at different timepoints can be evaluated [64].

According to (Figure 2), each sample presented a different behavior in the kinetic study of DPPH towards the extracts, and for the kinetic study of the ABTS assay, the extracts behaved similarly. As the measure of time than time increased, there was a decrease in absorbance and consequently a greater antioxidant capacity.

In the DPPH kinetics, the EMFMh showed a greater antioxidant effect during 50 min of testing; however, this behavior did not occur for the samples of EMGMh and EMRMh, as the antioxidant capacity was higher at the initial times between 10 min, and as time passed, the absorbance increased. For EMRMh, the best antioxidant capacity was around 20 min. After this time, the antioxidant capacity of the aqueous extract dropped again; however, it remained constant.

For the chemical kinetics of ABTS, both extracts showed strong antioxidant capacity compared to the chemical kinetics of DPPH; however, the EMFMh was the one with the best antioxidant capacity, as it obtained a lower absorbance at a lower concentration. Briefly, the EMFMh revealed better chemical kinetics, both in the DPPH and the ABTS assays.

Control experiments (without antioxidant standard), remained constant; however, the quercetin, rutin and gallic acid absorbances decreased over time, that is, the elimination of the DPPH radical occurred over time. Among the three standards, gallic acid was the one with the sharpest reduction in its absorbance and presented a better antioxidant capacity in relation to the others tested.

Similar results were reported by Magalhães et al. [65], in which there was a decrease in absorbance, which was proportional to the reaction time, for food samples in chemical kinetics by ABTS, with classic pattern (Trolox) and for a corresponding kinetic compound. Furthermore, when Trolox was added, constant absorbance values were reached after the first minute of measurement, indicating that the elimination reaction had been completed.

In a stopped-flow approach developed for monitoring the ABTS^•+^ reduction reaction by antioxidants, the determined total antioxidant capacity increases with increasing reaction time. According to the literature, structurally similar compounds have the same time-dependent behavior and pH, even if they have significant differences in the values of the total antioxidant capacity [66].

The variation in antioxidant capacity in different parts of the same plant may be due to the presence of some secondary metabolites with antioxidant action. For example, according to Simões et al. [67], generally, flavonoids found in leaves can be different from those present in flowers, branches, roots and fruits, the same compound can still present in different concentrations depending on the vegetable organ in which it is found.

### 2.5. Total Phenols and Total Flavonoids Content

Phenols are considered the molecules with the greatest potential to neutralize free radicals; these compounds act mainly as antioxidants due to their ability to scavenge free radicals and chelating metals in vitro and in vivo [68]. Plant phenolics include the presence of secondary metabolites such as flavonoids, condensed tannins, coumarins, and stilbenes [69].

Thus, the total phenolic content (TPC) of different plant extracts from leaves, branches, and roots of *M. hirsuta* measured according to the Folin-Ciocalteau method expressed in milligrams of equivalent gallic acid per gram of extract. Among the samples, EMGMh exhibited the highest phenolic content (1.07 ± 0.77 mg GAE/g), followed by EMFMh (0.58 ± 0.30 GAE/g) and (0.19 ± 0.47 mg GAE)/g) equivalent to the EMRMh.

For the content of total flavonoids (TFC) of different plant extracts EMFMh, EMGMh and EMRMh measured was measured spectrophotometrically using the colorimetric aluminium chloride assay. The highest TFC value was for EMFMh (5.12 ± 1.02 mg QR/g). For EMGMh and EMRMh, it resulted in the values (3.16 ± 0.88) and (2.04 ± 0.52) mg of equivalent rutin per gram of extract, respectively. The results of total phenols and total flavonoids indicated that all tested extracts had antioxidant capacity.

A previous study by Pereira et al. [15] determined the total phenolic content of the ethanol extracts from the leaves of *M. hirsuta* and found a TFC of (20.3 ± 0.08) μg EGA/mg. Another species of the genus Mansoa was reported by Abel et al. [70] and obtained similar results to our study, which reported the contents of total phenols and flavonoids in methanol extracts from the leaves of *Mansoa difficilis*, in which for the phenols it obtained (5.32 ± 0.01) mg EAG/g and total flavonoids (5.26 ± 0.17) mg EQ/g.

The phenolic content values varied slightly compared to those in the literature. The changes in the contents of total phenols and total flavonoids depend on several factors, such as the concentration of phenols and flavonoids, which depend on the polarity of the solvents used for extraction [71]. Environmental factors such as soil composition, temperature, rainfall, and incidence of ultraviolet radiation can affect the concentrations of phenolic compounds [72,73]. This may also be due to the presence of different amounts of sugars, carotenoids, ascorbic acid, or extraction methods that can change the amount of phenols [74].

The importance of evaluating the phenolic content in extracts is due to their scavenging capacity due to their hydroxyl groups [75]. Since there are reports that plant phenolics have many biological activities, including anticancer, antioxidant, and antimutagenic, the high consumption of phenolic compounds leads to a reduction in the risk of cardiovascular disease and cancer [76].

### 2.6. Determination of Antioxidant Activity Using Molecular Docking

To explore whether the established antioxidant activities of the triterpenes oleanolic acid (*m*/*z* 455 [M-H]^−^) and ursolic acid (*m*/*z* 455 [M-H]^−^) were identified as the main bioactive components of *M. hirsuta*, we undertook a molecular docking study. Three enzymes are used in this docking study: Tyrosinase, PRDX5 and Superoxide Dimutase (SOD1). Tyrosinase [77] is the main enzyme in the biosynthesis of melanin. Overproduction and accumulation of melanin occurs in several skin disorders. Since tyrosinase is the limiting step enzyme in melanogenesis, its inhibitors have become increasingly important as depigmenting agents in hyperpigmentation disorders. Currently, available tyrosinase inhibitors suffer from toxicity and/or a lack of efficacy and there is a constant need for better inhibitors from new natural sources as they are expected to be free of harmful side effects. PRDX5 [78] as well as SOD1 [79] have antioxidative and cytoprotective functions during oxidative stress. To gain an insight into the differences in binding between the compounds and these proteins, Rigid Receptor Docking (RRD) was performed.

Docking poses were analyzed and compared to the standard Quercetin. First, docking was performed on crystal structure of tyrosinase from *Bacillus megaterium* (PDB: 3NM8) [77]. Tyrosinase is a widely distributed copper-containing enzyme and is a member of the type 3 copper enzyme family that is involved in the production of melanin in a wide range of organisms. The second docking study was carried out on crystal structure of Human Peroxiredoxin 5, a Novel Type of *Mammalian Peroxiredoxin* (PDB: 1HD2) [78]. The peroxiredoxins define an emerging family of peroxidases able to reduce hydrogen peroxide and alkyl hydroperoxides with the use of reducing equivalents derived from thiol-containing donor molecules such as thioredoxin, glutathione, trypanothione and AhpF. Peroxiredoxins have been identified in prokaryotes as well as in eukaryotes. Peroxiredoxin 5 (PRDX5) is a novel type of mammalian thioredoxin peroxidase widely expressed in tissues and located cellularly to mitochondria, peroxisomes, and cytosol. Functionally, PRDX5 has been implicated in antioxidant protective mechanisms as well as in signal transduction in cells. The third docking study was performed on Human Superoxide Dismutase that (SOD1) protects cells from the effects of oxidative stress (PDB: 2C9V) [79]. Superoxide Dismutase (SOD) is an enzyme found in all living cells. Superoxide Dismutase helps break down potentially harmful oxygen molecules in cells. This might prevent damage to tissues.

The two triterpenes were subjected to docking analysis, and the specificities of their interaction with these targets, as shown in Figure 3, were investigated. Based on binding energies and interacting residues, the best-docked complexes were obtained. Docking poses were analyzed and compared to the standard antioxidant Quercetin. In all three molecular docking studies, Oleanolic acid and Ursolic acid docked very well compared to the standard Quercetin (Table 5, Figure 3).

Ligplots are shown in Figure 4. In both cases, Lys32 (B) was found to be involved in hydrogen bonding. Oleanolic acid interacts with tyrosinase, forming H-bonds at the receptor site interacting region involving residues Lys32(B) and Glu31(B) with distances of 3.29 Å and 3.16 Å respectively. Tyrosinase residues Arg263(A), Asp264(A), Phe258(A), Asn261(A), Pro257(A), Asn255(A) and Gly256(A) were involved in hydrophobic interactions. The interaction of ursolic acid with tyrosinase involves hydrogen bonding with Lys32(B) with distance of 3.07 Å. Hydrobhobic interactions of this compound were found with tyrosinase residues Arg263(A), Asp264(A), Phe258(A), Glu31(B), Gly256(A), Asn255(A), Asn261(A), and Pro25 7(A).

Ligplots in Figure 5 show that Oleanolic acid interacts with Human Peroxiredoxin 5 forming H-bond at the receptor site interacting region involving residue Arg124(A) with distance of 3.07 Å. Human Peroxiredoxin 5 residues Ala42(A), Thr44(A), Val180(A), Asn76(A) and Phe43(A) engaged in hydrophobic interactions. The hydrophobic interactions of ursolic acid with Human Peroxiredoxin 5 involves residues Arg124(A), Ala42(A), Val80(A), Phe43(A) and Glu83(A).

Ligplots show in Figure 6 depict Oleanolic acid interacting with Superoxide Dismutase forming an H-bond at the receptor site interacting region involving residue Gly108(A) with distance of 2.87 Å. Superoxide Dismutase residues Leu106(F), Ace21(F), Ile113(F), Ile151(F), Gly108(F), Cys111(A), Arg115(F), Leu106(A) and Ile151(A) were involved in hydrophobic interactions. Ursolic acid engaged in hydrogen bonding with Ile113(A) with distance of 2.77 Å. The hydrophobic interactions of Ursolic acid with Superoxide Dismutase involves residues Ile151(F), Ile113(F), Gly108(F), Ile151(A), Ser107(F), Cys111(A), Cys111(F), Leu106(A) and Ser107(A).

### 2.7. Pharmacophore Evaluation

Using the lowest energy conformers of Oleanolic acid and Ursolic acid, a pharmacophore model was generated [80]. The generated pharmacophore showed four key features: hydrogen bond acceptors (HBAs), hydrogen bond donors (HBDs), hydrophobic interactions (H), and Negative Ionizable Area (NI). The representative 3D and 2D pharmacophoric features of each compound are shown in Figure 7. Each compound constitutes individual pharmacophoric features and from these individual characteristic pharmacophores. A merged pharmacophore with common features was generated, as shown in Figure 8. This common feature pharmacophore model with a score of 0.9832 showed certain features: one HBD, three HBAs, seven Hs and one NI.

## 3. Materials and Methods

### 3.1. Chemicals and Equipment

The solvents used in the extraction and chromatographic analysis were of analytical and spectroscopic grade, including methanol, ethyl alcohol, acetic acid, and acetonitrile (Sigma Chemical Co., St. Louis, MO, USA). The extracts were concentrated in a *Fisatom* R-801 rotary evaporator, under reduced pressure with the aid of PRISMATEC BBV-132 vacuum pumps (Pfeiffer Vacuum, Aßlar, Germany).

For the chromatographic and spectrometric experiments, high purity HPLC grade solvents purchased from Sigma Aldrich^®^ (St. Louis, MO, USA) and ultra-pure water (18.2 MΩ in an Elga Purelab Option-Q system) were used. The stationary phase used was Silica C18 (particle size: 40-63 µm; Merck, Kenilworth, NJ, USA).

Antioxidant analyses were carried out using DPPH, ABTS, Folin-Ciocalteu reagent, gallic acid, Trolox, quercetin and rutin purchased from Sigma Chemical Co. (St. Louis, MO, USA).

In Silico Molecular Docking analyses were predicted and calculated using AutoDock Vina v.1.2.0 (The Scripps Research Institute, La Jolla, CA, USA).

### 3.2. Plant Sampling, Identification, and Extraction

Leaves, branches, and roots from *M. hirsuta* were collected in March and December 2020 at Sitio do Mocó, Coronel José Dias, near the municipality of São Raimundo Nonato-PI, (Northeast Brazil, geographic coordinates Latitude 09°00′55″ S × longitude 42°41′58″ W), under the registration number in the National System for the Management of Genetic Heritage and Associated Traditional Knowledge (SisGen) (A2CA781). After collection, leaves, branches, and roots were dried and ground at room temperature. Complete plant samples were sent to the Herbarium Graziela Barroso (TEPB) at UFPI, Teresina, PI, identified and deposited under voucher specimen (TEPB.32.277).

Different parts of the plant were crushed in an industrial blender and milled, and yielded 378.77 g of dried leaves, 650.12 g of dried branches and 107.32 of dried roots. The phytochemical extractions were carried out by maceration with methanol until exhaustion, followed by solvent evaporation in a rotary evaporator at 40 °C and 180 bar pressure.

### 3.3. Secondary Metabolites Content Assessment from M. hirsuta Extracts

#### 3.3.1. Phytochemical Screening

Preliminary phytochemical evaluation of each methanol extract was performed using colorimetric tests to detect the presence/absence of specific classes of phytochemical constituents (saponins, organic acids, phenols, tannins, flavonoids, and alkaloids) [81].

Comparative Thin Layer Chromatography analysis using the following solvent systems: ethanol/ethyl acetate 8:2 (*v*/*v*); ethyl acetate/methanol 8:2 (*v*/*v*); ethyl acetate/methanol 7:3 (*v*/*v*); ethyl acetate/methanol 9:1 (*v/v*), hexane/ethyl acetate 8:2 (*v*/*v*); Chloroform/methanol 9:1 (*v*/*v*). The method was carried out using 60778-25EA silica gel thin layer chromatoplates, fluorescing at 254 nm (20 cm × 20 cm, 0.25 mm thick) obtained from Sigma-Aldrich. The plates were developed by UV irradiation (254 nm and 366 nm) and with physical developer ceric sulfate. The results were expressed as the intensity of staining after developing the chromatoplate.

#### 3.3.2. Determination of Chemical Composition by HPLC-PDA, LC-MS, and ^1^H-NMR

Concerning the HPLC-PDA analysis, the methanol extracts of leaves, branches, and roots of *M. hirsuta* were analyzed at room temperature using a Shimadzu analytical liquid chromatograph (Shimadzu, Kyoto, Japan), model LC20A, CBM-20 controller, UV-visible detector with “Diode Array” (DAD) model SPDM-20A, DGU-20A3 degasser and LC solutions software (Shimadzu Corporation, Kyoto, Japan). A quantity of 1 mg of *M. hirsuta* extracts was injected, using a gradient method up to 50 min, and a mobile phase of a mixture of water + 0.1% acetic acid (A) + 100% acetonitrile (B), which were pumped at a rate 1.0 mL min^−1^, Stationary phase C18 column reversed phase (250 × 4.6 mm, particle size 5 µm), brand Macherey-Nagel (Düren, Germany) and membrane filtered 0.45 µm, the column oven was heated to 40 °C using a UV-visible detector chromatograph in 50 min.

LC-MS analysis of the extracts were performed using a Shimadzu^®^ LC System equipment, LC-20AD pump, coupled to SPD-M20A Diode Array detector, and automatic injector (model SIL-20AHT), Communication module: CBM-20ª, Auto-MS mode. MS/MS, frag Ampl: 75%. Full: 50–1200 *m*/*z*. Mass Spectrometer: Amazon SL Bruker^®^ (Billerica, MA, EUA). Ionization source: electrospray (ESI). Analyzer: ionTrap. HPLC method. The mobile phase was programmed in a mixture of methanol B and ultra-pure water A solvents. Exploratory gradient: 5–100% B (45 min), 100% B (45–55 min), 5% B (55–57 min) and from (57–68 min) remaining at 5% to re-equilibrate the column at a flow rate of 1 mL min^−1^. As stationary phase, a C-18 Phenomenex Gemini column (250 × 4.60 mm; 5 m; 110 Å) was used in the analytical mode, with extract concentration 1 mg mL^−1^.

One-dimensional NMR spectra were obtained on a Bruker Avance 600 spectrometer (Billerica, MA, USA) with a 5 mm TCI cryoprobe and a 14.1 T magnetic field, operating at 600 MHz for 1H. ^1^H chemical shifts were referenced to the DMSO-*d*_6_ solvent peak (δ 2.49, Cambridge Isotope Laboratories, Inc., Tewksbury, MA, USA), used to solubilize the samples.

### 3.4. Antioxidant Activity Assessment

#### 3.4.1. Antiradical Potential towards DPPH^•^ (2,2-Diphenyl-1-picrylhydrazyl)

Serial dilutions of each sample were prepared. 1.8 mL of DPPH solution (0.06 mM) were added to 200 μL of each sample at different concentrations (60, 70, 80, 90 and 100 μg mL^−1^). After 20 min from the start of the reaction in the dark, the absorbance of each sample was measured at 517 nm using a UV-Vis spectrophotometer (model: SP-220, Biospectro). The determinations were made using a negative control (DPPH solution with methanol, blank) and a positive control (quercetin) [82]. The results were expressed as the total percentage of antioxidant activity (AA%) and the IC_50_ parameter. IC_50_ values denote the concentration of the sample, which is required to scavenge 50% of DPPH free radicals Equation (1):(1)$$\mathrm{AA}=\left[\;\frac{\mathrm{Abscontrol}-\left(\mathrm{Abssample}-\mathrm{Absblank}\right)}{\mathrm{Abscontrol}}\right]\times 100$$
where: Control Abs is the control absorbance, Sample Abs is the test sample absorbance and Blank Abs is the blank absorbance.

From the percentage of inhibition, the concentration effective to inhibit 50% of the DPPH^•^ radical (IC_50_) was estimated using a simple linear regression model.

#### 3.4.2. Antiradical Potential towards ABTS^•+^ (2,2′-Azino-bis (3-ethylbenzothiazoline-6-sulfonic Acid)

Twenty-microliter aliquots of different concentrations of samples of EMFMh (10–50 μg mL^−1^), EMGMh (10–70 μg mL^−1^) and EMRMh (30–80 μg mL^−1^) extracts, were mixed with 1.8 mL of ABTS^•+^ solution on an ELISA plate. After 6 min of dark reaction, the decrease in absorbance was measured at 734 nm. Trolox was used as a positive control. Determinations were carried out in triplicates. The ABTS^•+^ radical scavenging activity was expressed as a percentage using the same formula as the DPPH assay (Equation (1)). From the percentage inhibition obtained for the samples of methanol extracts from leaves, branches and roots, the mean inhibitory concentrations (IC_50_, concentration of the sample required to reduce the initial ABTS concentration by 50%) in μg. mL^−1^ were determined [83].

#### 3.4.3. DPPH and ABTS Chemical Kinetics

Similar to the DPPH method, in an Elisa plate, 1.8 mL of DPPH solution (0.06 mM) was added to 200 μL of each extract, referring to EMFMh, EMGMh, and EMRMh in a concentration of 100 μg mL^−1^. At the end of the reading, the % DPPH (Y axis) vs. time (X axis) results were plotted in graphs to generate kinetic curves [84]. The determinations were carried out using a negative control (DPPH solution with methanol) and positive control (quercetin).

For ABTS, 20 μL aliquots of different concentrations of samples from EMFMh (50 μg mL^−1^), EMGMh (70 μg mL^−1^), and EMRMh (80 μg mL^−1^) were mixed with 1.8 mL of ABTS^•+^ (7 mM) solution in an Elisa board. At 6, 10, 20, 30 and 60 min the absorbance was measured at 734 nm in a spectrophotometer in the dark [85]. At the end of the reading, the % ABTS (Y axis) vs. time (X axis) results were plotted in graphs to generate kinetic curves. The determinations were carried out using a negative control (ABTS solution with ethanol) and a positive control (trolox).

#### 3.4.4. Determination of Total Phenols Content

The total phenolic content of the extract was determined with a standard curve method using gallic acid at different concentrations (20–100 μg μL^−1^). For the reaction, 2 mL of distilled water was mixed with 250 μL of Folin Ciocalteu reagent and 250 μL of the extracts. After the light protection period (8 min), 100 μL of sodium carbonate solution (10% *w*/*v*) was added. The solutions were mixed and allowed to stand in the dark for 1 h at room temperature. Absorbance was measured at 760 nm using a UV-Vis spectrophotometer. To determine the total phenolic content, gallic acid was used to make the standard calibration curve (10–160 μg μL^−1^). The absorbance of the samples was applied to the standard line equation (Y = 0.001569x + 0.01405, r^2^ = 0.9939), where y = absorbance and x = gallic acid concentration (Appendix A in the Appendix A). The final concentration of the total phenolic content present in the extracts was expressed in mg of Gallic Acid Equivalents (EAG) per g of dry weight of extract [86].

#### 3.4.5. Total Flavonoid Content

A volume of 1000 μL of each sample (500 μg mL^−1^ diluted in methanol) was added to 1000 μL of 2% Aluminum Chloride (AlCl_3_) (diluted in methanol). The solutions were mixed and placed under light protection at room temperature (25 °C) for 1 h. The absorbance was measured at 420 nm using a UV-Vis spectrophotometer: SP-220, Biospectro (Analítica^®^, São Paulo, Brazil). To determine the total flavonoids content, a calibration curve was constructed (Y = 0.459x − 0.2686; r^2^ = 0.9943) with the standard rutin (10–160 μg mL^−1^). The total flavonoid content was expressed in mg of rutin equivalent (ER)/g per gram of extract dry weight [87].

### 3.5. Statistical Analysis

Statistical analysis for antioxidant tests was performed in triplicate and data presented as means ± standard deviations (SD) as IC_50_ values, using Graph Pad Prism 6.0 (Graph Pad Prism Software Inc., San Diego, CA, USA).

### 3.6. Molecular Docking

Molecular docking analysis was performed using Autodock Vina v.1.2.0 (The Scripps Research Institute, La Jolla, CA, USA) docking software [88].

The receptor site was predicted using LigandScout (Inte: Ligand) Advanced software [80] (evaluation license key: 81809629175371877209), which identifies putative binding pockets by creating a grid surface and calculating the buriedness value of each grid point on the surface.

The resulting pocket grid consists of several clusters of grid points, rendered using an iso surface connecting the grid points to each other. The iso surface represents empty space that may be suitable for creating a pocket. The x-ray crystal structure of Tyrosinase from *Bacillus megatarium* (PDB: 3NM8) [77], Human Peroxiredoxin 5, a Novel Type of Mammalian Peroxiredoxin (PDB: 1HD2) [78], and Atomic resolution structure of Cu-Zn Human Superoxide Dismutase (PDB: 2C9V) [79] were retrieved from the Protein Data Bank and utilized to perform docking simulations.

The box center and size coordinates for (PDB: 3NM8) was −5.59208 Å × −3.33416 Å × 10.2111 Å and 31.5668 Å × 22.6662 Å × 23.0593 Å; for (PDB: 1HD2) was 18.9015 Å × 44.192 Å × 28.1802 Å and 12.81 Å × 17.1255 Å × 15.8749 Å; for (PDB: 2C9V) was 31.8729 Å × −0.108084 Å × 14.3963 Å and 13.8504 Å × 18.6613 Å × 35.181 Å around the active site.

Default search parameters were used where number of binding modes were 10, exhaustiveness was 8, and maximum energy difference was 3 kcal/mol. Chimera 1.16, UCSF_USA [89] LigPlot+ software (EMBL-EBI, Cambrigeshire, UK) [90] and Samson 2022 (OneAngstrom, French Institute for Research in Computer Science and Automation, Domaine de Voluceau, Frence), [91] were used for visualization and calculation of protein–ligand interactions.

### 3.7. 3D Pharmacophore Model Generation

LigandScout by Inte Ligand, Advanced software (Wolber and Langer), Vienna, Austria, Europe [80] (evaluation license key: 81809629175371877209) was used to generate a 3D pharmacophore model. Espresso algorithm was used to generate ligand-based pharmacophore. The generated pharmacophore model compatibility with the pharmacophore hypothesis was created using default settings for LigandScout. Relative Pharmacophore-Fit scoring function, Merged feature pharmacophore type and feature tolerance scale factor was set to 1.0 for Ligand-Based Pharmacophore creation. The best model was selected from the 10 generated models.

## 4. Conclusions

This research demonstrates that the antioxidant capacity is altered for different parts of the same plant, as in the detection method, which means that the antioxidant compounds present in this plant are probably different in structure and quantity, as the data from this study showed a better antioxidant capacity present in the leaves (EMFMh), followed by branches (EMGMh) and roots (EMRMh). These results also show the importance of selecting several methods to quantify the antioxidant activity of plant extracts. The presence of different phytochemicals observed in chemical profiles may justify pharmacological activities, such as their antioxidant potential. Also, an evaluation of the antioxidant activities of the triterpenes oleanolic acid and ursolic acid was identified as the main bioactive components of *M. hirsuta*. Molecular docking analysis was performed on the two triterpenes to determine whether the compounds bound to the important receptors which are connected to antioxidant mechanisms. These compounds gave good binding potentials associated with antioxidant activity. From these results, a pharmacophore model was proposed to help guide future studies. Additionally, the proposed pharmacophore model should be used as a future guide for selecting and designing triterpenes as antioxidants. Further research is required to assess the toxicity and elucidate its mechanism of action.

## Figures and Tables

**Figure 1 molecules-27-06016-f001:**
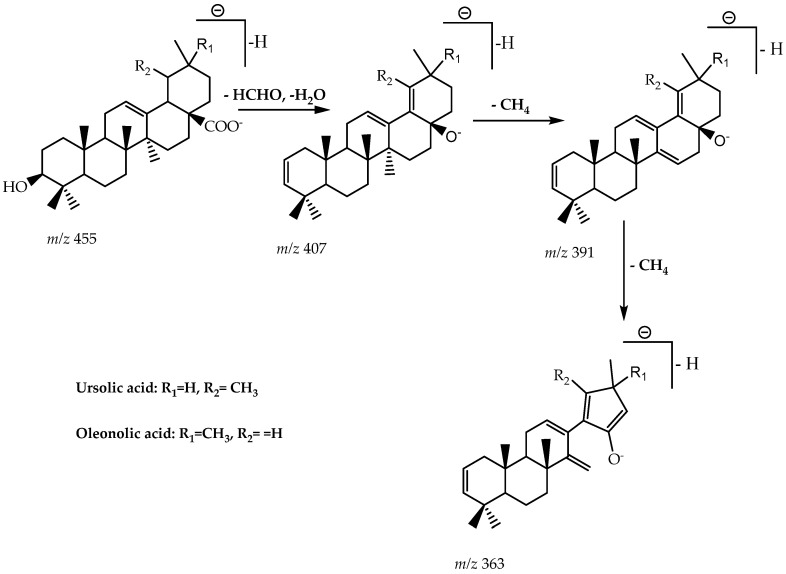
The proposed fragmentation pathways of Oleonolic acid and Ursolic acid.

**Figure 2 molecules-27-06016-f002:**
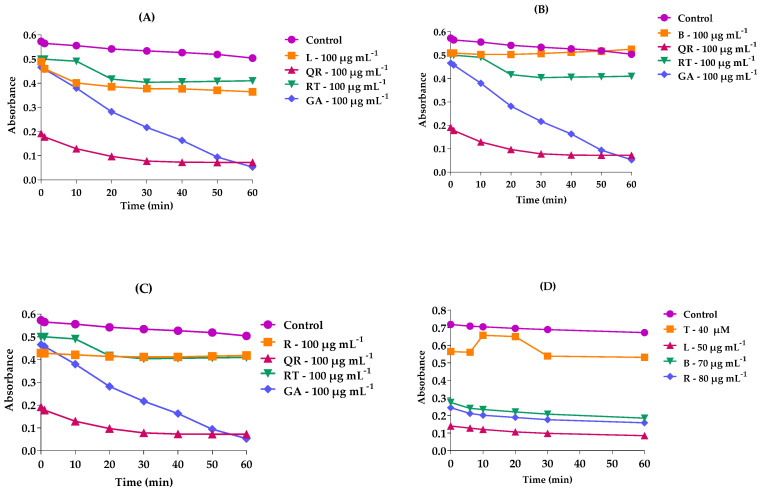
Kinetic curve of the antioxidant capacity of *M. hirsuta* from (**A**) DPPH—methanol extract of leaves, (**B**) DPPH—methanol extract of branches, (**C**) DPPH—methanol extract of roots, compared to rutin, quercetin, gallic acid as standards, and (Control—no antioxidant). (**D**) ABTS—methanol extracts of leaves, branches and roots compared to TROLOX (40 μM) and control (without antioxidant). L—Leaves, B—Branches, R—Roots; positive controls: RT: rutin, QT: quercetin and GA: gallic acid.

**Figure 3 molecules-27-06016-f003:**
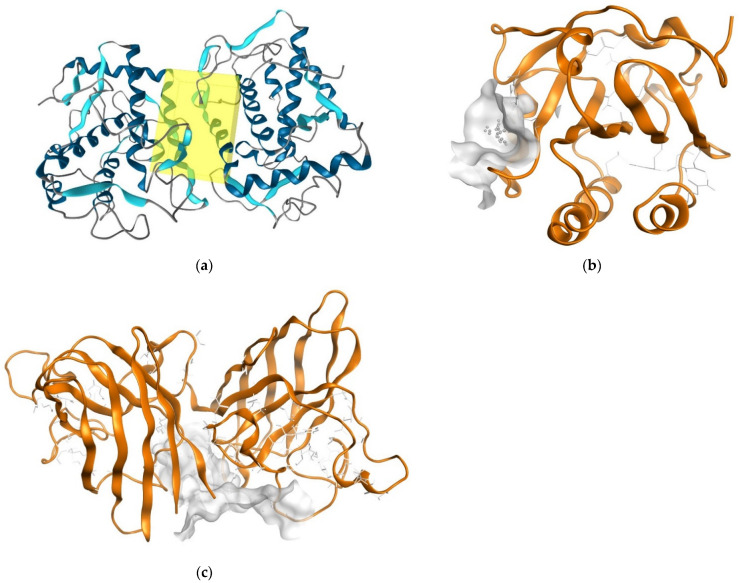
Binding site (yellow color) of tyrosinase from *Bacillus megaterium* (**a**); Binding site (grey-white color) of Human Peroxiredoxin 5 (**b**); Binding site (grey-white color) of Superoxide Dismutase (SOD1) (**c**).

**Figure 4 molecules-27-06016-f004:**
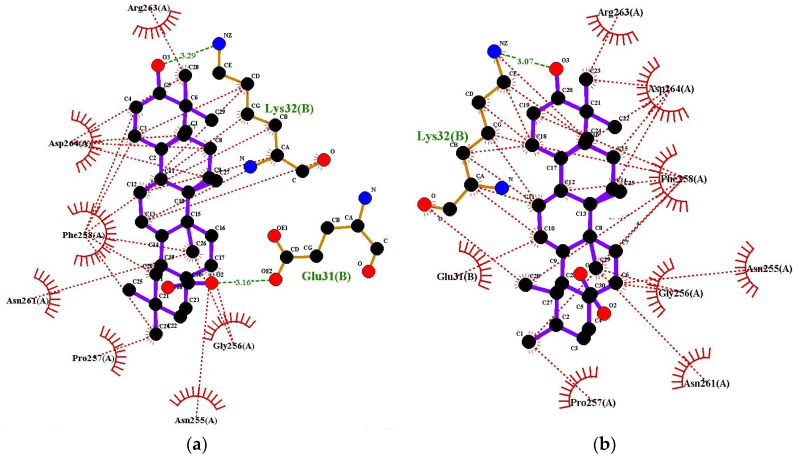
Ligplots showing interacting residues of Tyrosinase complex with Oleanolic Acid (**a**) and Ursolic Acid (**b**). Purple lines, triterpenes ligands; green dotted lines, hydrogen bonds labelled with distances in Å; red dotted lines, hydrophobic interactions; red circles, oxygen atoms; blue circles, nitrogen atoms.

**Figure 5 molecules-27-06016-f005:**
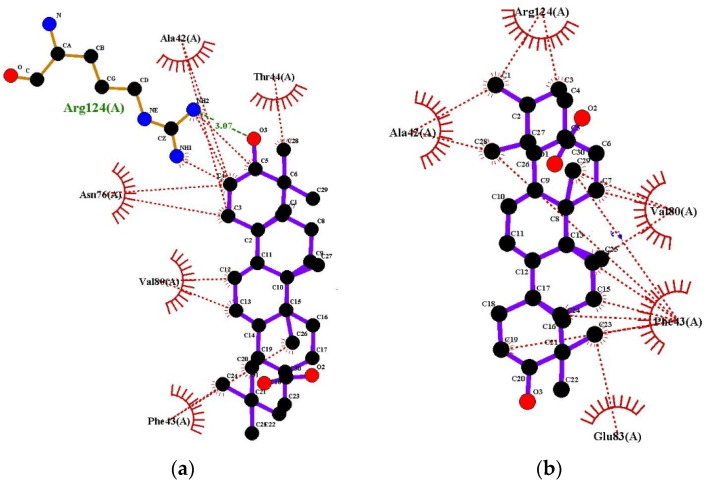
Ligplots showing interacting residues of Human Peroxiredoxin 5 in complex with Oleanolic Acid (**a**) and Ursolic Acid (**b**). Purple lines, triterpenes ligands; green dotted lines, hydrogen bonds labelled with distances in Å; red dotted lines, hydrophobic interactions; red circles, oxygen atoms; blue circles, nitrogen atoms.

**Figure 6 molecules-27-06016-f006:**
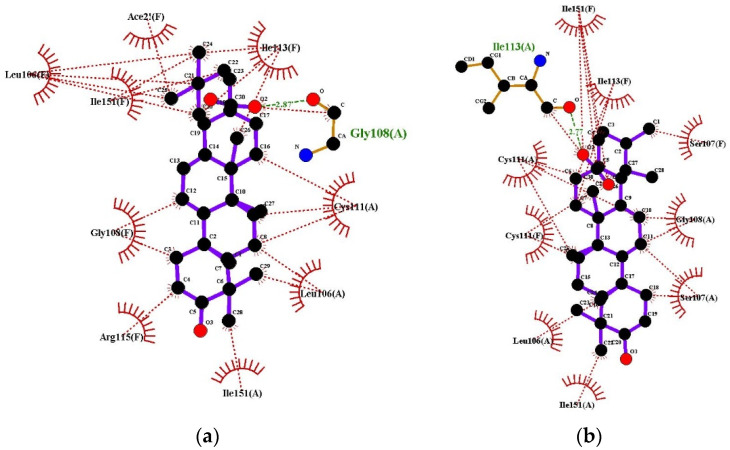
Ligplots showing interacting residues of Superoxide Dismutase (SOD1) in complex with Oleanolic Acid (**a**) and Ursolic Acid (**b**). Purple lines, triterpenes ligands; green dotted lines, hydrogen bonds labelled with distances in Å; red dotted lines, hydrophobic interactions; red circles, oxygen atoms; blue circles, nitrogen atoms.

**Figure 7 molecules-27-06016-f007:**
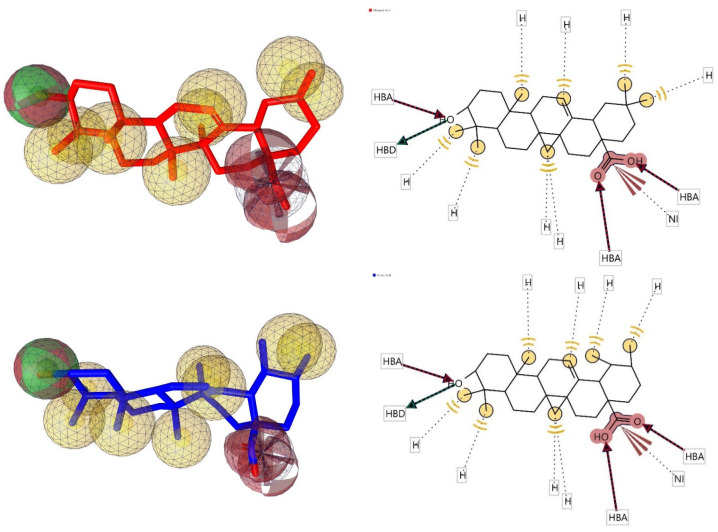
3D and 2D representations of pharmacophoric features of Oleanolic acid and Ursolic acid used in 3D pharmacophore generation. Red, HBAs; green, HBDs; Yellow, H; Brown, NI as described earlier.

**Figure 8 molecules-27-06016-f008:**
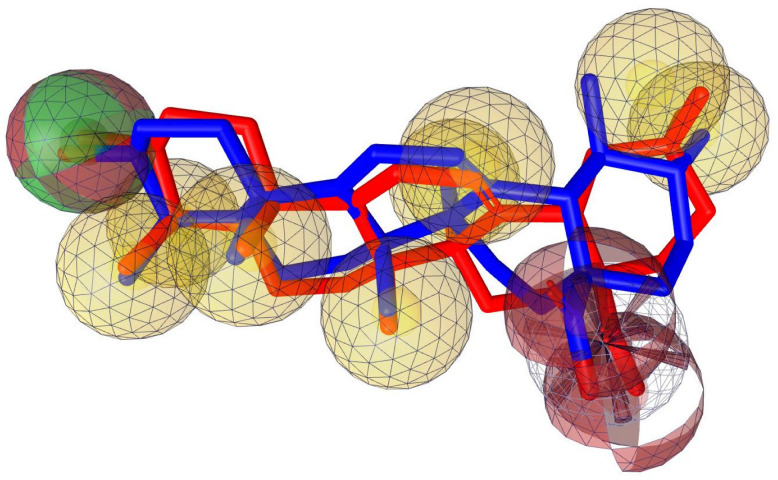
Common feature pharmacophore. Color codes analogous to Figure 7 Red, HBAs; green, HBDs; Yellow, H; Brown, NI.

**Table 1 molecules-27-06016-t001:** Classes of secondary metabolites identified in methanol extracts from leaves, branches, and roots of *M. hirsuta*.

Saponins		Organic Acids	Phenols and Tannins	Flavonoids	Alkaloids	Catechins
EMFMh	+	+	+	-	+	-
EMGMh	-	+	+	+	+	-
EMRMh	-	+	+	-	+	-

(+) Presence; (-) Absence of compounds.

**Table 2 molecules-27-06016-t002:** Identification by HPLC-PDA of compounds from *M. hirsuta* extracts UV-Visible Absorptions.

*M. hirsuta*	R_t_ (Min)	λ Max	Classes of Compounds	Reference
EMFMh, EMGMh and EMRMh	2.69	260/269	Flavonoids (isoflavone)	[31]
	17.24	273	Proanthocyanidins	[32,33]
	18.27	284/350		
	19.49	282/350		
	20.70	285/327		
	21.18	204/283	Flavonoids (flavanones or dihydroflavonols)	[31]
	21.24	214	Triterpenes (Oleanolic acid)	[34]
	22.65	214	Triterpenes (Ursolic acid)	[34]

EMFMh: Methanol Extract of *Mansoa hirsuta* leaves; EMGMh: Methanol Extract of *Mansoa hirsuta* branches; EMRMh: Methanol Extract of *Mansoa hirsuta* roots.

**Table 3 molecules-27-06016-t003:** Analysis of extracts of *M. hirsuta* leaves, branches, and roots by LC-MS.

Extract	R_t_ (Min)	Ionization Mode	Fragments/% Abundance	Compound	Reference
EMFMh	43.9	[M-H]—negative	*m*/*z* 363 (4.4%), 407 (3.4%), *m*/*z* 455 (0.8%)	Oleanolic acid and Ursolic acid	[40,41]
EMGMh	45.6	[M-H]—negative	*m*/*z* 363 (19.4%), 391 (5.6%), *m*/*z* 407 (16.7%)	Oleanolic acid and Ursolic acid	[41]
EMRMh	27.0	[M-H]—negative	*m*/*z* 363 (29.1%), 407 (16.7%), *m*/*z* 455 (5.8%)	Oleanolic acid and Ursolic acid	[40,41]

R_t_ = Retention time.

**Table 4 molecules-27-06016-t004:** Antiradical activity towards DPPH and ABTS of methanol extracts of *M. hirsuta* (mean ± standard deviation).

Samples	DPPH (µg mL^−1^)—IC_50_	ABTS (µg mL^−1^)—IC_50_
EMFMh	78.9 ± 0.05	43.5 ± 0.14
EMGMh	77.3 ± 0.03	63.6 ± 0.54
EMRMh	82.7 ± 0.1	56.1 ± 0.05
Quercetin	41.0	-
Trolox	-	73.2

EMFMh: Methanol Extract of *Mansoa hirsuta* leaves; EMGMh: Methanol Extract of *Mansoa hirsuta* branches; EMRMh: Methanol Extract of *Mansoa hirsuta* roots.

**Table 5 molecules-27-06016-t005:** Docking analysis of two triterpenes ligands on three different protein receptors with respect to Quercetin standard.

Compounds	Docking Score (-) (kcal/mol)	Docking Score (-) (kcal/mol)	Docking Score (-) (kcal/mol)
	PDB ID: 3NM8 (Bacterial Tyrosinase)	PDB ID: 1HD2 (Human Peroxiredoxin 5)	PDB ID: 2C9V (Human Superoxide Dismutase)
Quercetin (Standard)	9.2	3.3	7.1
Oleanolic Acid	8.3	6.4	8.6
Ursolic Acid	8.2	6.3	8.9

## Data Availability

Not applicable.

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
