# Peer review of "The Free Radical Scavenging Property of the Leaves, Branches, and Roots of Mansoa hirsuta DC: In Vitro Assessment, 3D Pharmacophore, and Molecular Docking Study"

_molecules, 2022, doi:10.3390/molecules27186016_

Round 1
Reviewer 1 Report
In this study, the authors examined the antioxidant potential of extracts obtained from different parts of the Mansoa hirsuta plant (leaves, branches, and roots). For this purpose, DPPH and ABTS tests were performed, as well as determination of total phenols and total flavonoids content. In addition, molecular docking analysis were performed in order to better understand the potential mechanism of the antioxidant activity of the major metabolites.
This paper is interesting, but it is necessary to make some corrections:
- - Abbreviations for extracts (EMFMh: Methanol Extract of Mansoa hirsuta leaves; EMGMh: Methanol Extract of Mansoa hirsuta branches; EMFMh: Methanol Extract of Mansoa hirsuta roots) are given below Table 4, and appear in the text before this table. Also, when specifying these abbreviations, a mistake was made with the abbreviation for root extract. Therefore, the abbreviation EMFMh appears twice. I suggest that these abbreviations be introduced in the text where they are first mentioned.
- - Folin-Ciocalteu method was used to determine total phenolic content in different plant extracts from leaves, branches and roots of M. hirsuta. In part of the manuscript 3.4.4. Determination of Total Phenols Content, the procedure is not described. This procedure should be briefly described. The authors expressed the results in milligrams of gallic acid equivalents per gram of extract. I suggest to show the obtained standard curve in the Manuscript or Supplementary material.
- Docking poses were analyzed and compared with the co-crystallized standard antioxidant nordihydroguaiaretic acid (NDGA), but in the experiment quercetin was used as a standard for the DPPH test. I suggest that an in vitro experiment be done with NDGA.
Author Response
Editor, Molecules
I forward the manuscript, revised and entitled “Free radical scavenging property of the leaves, branches, and roots of Mansoa hirsuta DC: In vitro assessment, 3D pharma-cophore and molecular docking study", (molecules-1920018) by Patrícia e S. Alves, Gagan Preet, Leandro Dias, Maria Oliveira, Rafael Silva, Isione Castro, Giovanna Silva, Joaquim Júnior, Nerilson Lima, Dulce Helena Silva, Teresinha Andrade, Marcel Jaspars, and Chistiane Feitosa for review in your journal.
Follows letter containing response to items submitted by reviewers.
It is noteworthy that the authors agree with all points raised, which have been incorporated in this new version. We took the opportunity to thank the reviewers, as these adjustments and corrections significantly improved the quality of the article.
As suggested the changes made in the manuscripts are in red.
Comments to the Author
Reviewer: 1
Abbreviations for extracts (EMFMh: Methanol Extract of Mansoa hirsuta leaves; EMGMh: Methanol Extract of Mansoa hirsuta branches; EMFMh: Methanol Extract of Mansoa hirsuta roots) are given below Table 4, and appear in the text before this table. Also, when specifying these abbreviations, a mistake was made with the abbreviation for root extract. Therefore, the abbreviation EMFMh appears twice. I suggest that these abbreviations be introduced in the text where they are first mentioned.
Done. The items have been corrected.
2.Folin-Ciocalteu method was used to determine total phenolic content in different plant extracts from leaves, branches and roots of M. hirsuta. In part of the manuscript 3.4.4. Determination of Total Phenols Content, the procedure is not described. This procedure should be briefly described. The authors expressed the results in milligrams of gallic acid equivalents per gram of extract. I suggest to show the obtained standard curve in the Manuscript or Supplementary material.
Done. The procedure was described and Supplementary Material in Figure S5 show the Standard curve of total phenols from M. hirsuta extracts
3 Docking poses were analyzed and compared with the co-crystallized standard antioxidant nordihydroguaiaretic acid (NDGA), but in the experiment, quercetin was used as a standard for the DPPH test. I suggest that an in vitro experiment be done with NDGA.
Done. Docking results have been changed taking quercetin as standard

Reviewer 2 Report
The work presented here by Alves is very interesting and well presented. The experimental methodology followed is consistent and the results are encouraging. Nevertheless, some adjustments should be made to improve the overall quality of the paper.
1. Page 2 line 88: the abbreviation DPPH and ABTS should be written in full as this is the first time they are mentioned.
2. page 11 lines 391-393: The Ramachandran plot is not used to determine the stereochemistry of a protein, but rather the secondary structure of the amino acidic sequence and the energetically-favoured combinations of the dihedral angles. This should be corrected. Moreover, I see no reason to add Ramachandran plots as the structures used are deposited, and, therefore, already validated, in the Protein Data Bank.
3. Page 11 lines 397-399: for the reasons explained above, those lines should be removed.
4. Page 11 lines 400-404: The corresponding reference should be added after every PDB structure listed.
5. Page 13 line 443: "Figure 5" should not be in parentheses.
6. Page 15 line 486: Though it is reported in the Materials and Methods section, authors should indicate the software used for the development of the pharmacophore here too.
7. Pag 16 line 499: "Figure xyz"
8. Pag 19 line 645: coordinates have no unit, only the box dimensions are in Angstrom.
Author Response
Editor, Molecules
I forward the manuscript, revised and entitled “Free radical scavenging property of the leaves, branches, and roots of Mansoa hirsuta DC: In vitro assessment, 3D pharma-cophore and molecular docking study", (molecules-1920018) by Patrícia e S. Alves, Gagan Preet, Leandro Dias, Maria Oliveira, Rafael Silva, Isione Castro, Giovanna Silva, Joaquim Júnior, Nerilson Lima, Dulce Helena Silva, Teresinha Andrade, Marcel Jaspars, and Chistiane Feitosa for review in your journal.
Follows letter containing response to items submitted by reviewers.
It is noteworthy that the authors agree with all points raised, which have been incorporated in this new version. We took the opportunity to thank the reviewers, as these adjustments and corrections significantly improved the quality of the article.
As suggested the changes made in the manuscripts are in red.
Reviewer: 2
Comments to the Author
- Page 2 line 88: the abbreviation DPPH and ABTS should be written in full as this is the first time they are mentioned
We made changes as suggested.
page 11 lines 391-393: The Ramachandran plot is not used to determine the stereochemistry of a protein, but rather the secondary structure of the amino acidic sequence and the energetically-favoured combinations of the dihedral angles. This should be corrected. Moreover, I see no reason to add Ramachandran plots as the structures used are deposited, and, therefore, already validated, in the Protein Data Bank.
Done. Ramachandran's text and plot figure has been removed. I have renumbered the others figures.
- Page 11 lines 397-399: for the reasons explained above, those lines should be removed.
Done. Ramachandran's text and plot figure has been removed. I have renumbered the others figures.
Page 11 lines 400-404: The corresponding reference should be added after every PDB structure listed.
Done. I have added one more software reference no. 92 in the manuscript
Page 13 line 443: "Figure 5" should not be in parentheses.
Done. The item has been corrected in the revised version
- Page 15 line 486: Though it is reported in the Materials and Methods section, authors should indicate the software used for the development of the pharmacophore here too.
Done. I have added
- Pag 16 line 499: "Figure xyz"
Done. The item has been corrected in the revised version
- Pag 19 line 645: coordinates have no unit, only the box dimensions are in Angstrom.
We have made changes as suggested.
